

# Examination of distraction and discomfort caused by using glare monitors: a simultaneous electroencephalography and eye-tracking study

Yoritaka Akimoto and Keito Miyake

Department of Information and Management Systems Engineering, Nagaoka University of Technology, Nagaoka, Japan

## ABSTRACT

**Background:** Since the COVID-19 pandemic started, remote work and education and digital display use have become more prevalent. However, compared with printed material, digital displays cause more eye fatigue and may decrease task performance. For instance, the reflections on the monitor can cause discomfort or distraction, particularly when glare monitors are used with black backgrounds.
**Methods:** This study simultaneously uses electroencephalography (EEG) and an eye-tracker to measure the possible negative effects of using a glare monitor on the illegibility of sentences.
**Results:** The experiment results showed no difference in reading time and subjective illegibility rating between glare and non-glare monitors. However, with glare monitors, eye fixation when reading lasted longer. Further, EEG beta (15–20 Hz) power variations suggested that the participants were less engaged in the reading task when a glare monitor was used with a black background.
**Conclusions:** These results indicate that the negative effects of using a glare monitor are subtle but certainly present. They also show that physiological measures such as EEG and eye tracking can assess the subtle effects in an objective manner, even if behavioral measures such as subjective illegibility ratings or reading time may not show the differences.

# INTRODUCTION

The need for literacy in handling computers and other digital devices is increasing yearly, and digital devices are expected to become more widespread in the future. In addition, the COVID-19 situation, which started in the end of 2019, dramatically popularized remote work and education using digital devices. However, digital screens also pose problems; for instance, they are more tiring than printed materials, which could lead to asthenopia or reduced task performance (*Rosenfield, 2011*; *Yan et al., 2008*).

Glare, which is a phenomenon that interferes with good visibility and can cause disabilities (disability glare) or discomfort (discomfort glare) (*Van Den Berg, 1991*), makes

Corresponding author
Yoritaka Akimoto,
y-akimoto@kjs.nagaokaut.ac.jp

reading using a digital screen more difficult and tiring. The effect of discomfort glare is the main concern when using a digital screen. Two types of monitors are commonly used in these screens: a non-glare monitor, which has no gloss and does not reflect light easily, and a glare monitor, which has gloss and reflects light. Subjectively, the glare monitor reduces task performance because of the discomfort or distraction caused when the surrounding environment is reflected. This effect appears to be prominent when the background color is black, as this causes reflections to stand out in the glare monitor screens. However, to the best of our knowledge, no study has used an eye tracker or electroencephalography (EEG) to measure and compare objective physiological responses when using glare and non-glare monitors. Meanwhile, both physiological methods have been used to examine cognitive processing during text comprehension (*Raney, Campbell & Bovee, 2014*; *Ditman, Holcomb & Kuperberg, 2007*) and assess cognitive effort (*Zhu, Wang & Zhang, 2021*).

Although no previous study has directly examined the effects of using a glare monitor, the effects of glare on task performance have been studied. For example, *Huang & Menozzi (2014)* examined the effect of discomfort glare on detecting and processing visual information. In their study, discomfort glare was introduced by illuminating a blank frame prior to target presentation. They found that discomfort glare impaired visual performance in the peripheral visual field. Further, *Ko et al. (2014)* investigated the effect of discomfort glare on task performance and viewing distance. This cited study included text-based visual search and matching tasks, and discomfort glare was introduced using light-emitting diode (LED) light reflected off a matte liquid crystal monitor that contained some text. Results showed that adding reflective glare did not reduce task performance, albeit viewing distance was reduced. These findings suggest that, compared to central vision, peripheral vision is more susceptible to glare.

Previous studies have also examined the effect of glare on physiological responses. *Hamedani et al. (2019)* reviewed the methods used to measure physiological responses to perceived glare, especially using eye-tracking technology. They concluded that relative pupil size was the most promising eye-tracking measure; eye movement was considered as the second most promising measure, but it required further investigation. *Bargary et al. (2015)* used functional magnetic resonance imaging to investigate the neural mechanisms associated with discomfort glare. They measured the participants' sensitivity to discomfort glare and compared the brain activity between a sensitive group and a less sensitive group based on three levels of glare intensity. In their study, discomfort glare was introduced using four LED lights mounted around the target stimuli. Results showed that the sensitive group experienced more activation in visual areas than the less sensitive group did, regardless of the discomfort glare level.

*Muto, Munakata & Sano (2021)* also examined the effect of discomfort glare on brain activity using EEG. In their study, discomfort glare was introduced by manipulating the luminance of a light-emitting device, which was installed within the frame of the pseudo window. They compared the alpha (8–13 Hz) and beta (13–30 Hz) powers between each luminance condition—with and without performing a calculation task. The EEG

frequency powers obtained from one minute of open-eye resting state at the beginning of the experiment were considered as baseline. *Muto, Munakata & Sano (2021)* found the main effect of luminance in alpha power in the O1 channel, significant interactions between luminance and task factors in beta power in the O1 channel, and between alpha and beta powers in the Fp1 and Fp2 channels. However, whether the observed effects were related to discomfort glare was not clear; specifically, since the main luminance effect was observed regardless of whether a task was being performed, it may be that this effect simply reflects the difference in luminance. It was also unlikely that the significant interaction effect observed in the alpha and beta powers in the frontal channel could be related to discomfort glare because the effect was mainly caused by the low-luminance-without-task condition. Conversely, the significant interaction observed in beta power in the O1 channel might be related to discomfort glare because this interaction was caused by a high luminance 20,000 cd/m$^2$ condition with a low legibility rate. The authors interpreted this result as decreased visual processing under the high luminance 20,000 cd/m$^2$ condition.

In the present study, we used EEG and an eye tracker to examine whether the possible negative effects of reading text on a glare monitor could be measured using objective physiological responses. Specifically, theta, alpha, and beta powers when reading sentences, as well as the reading time and eye-tracking data, were compared when using a glare and a non-glare monitor. Regarding eye-tracking data, we analyzed the fixation counts and fixation duration, instead of relative pupil dilation. This was done despite the latter having been associated with cognitive effort (*van der Wel & Van Steenbergen, 2018*) and considered as the most promising eye-tracking measure for assessing perceived glare (*Hamedani et al., 2019*). The reasoning behind the decision is that pupil size is mostly determined by the background luminance; therefore, utilizing it in the present study would be difficult.

We also examined the correlations between EEG frequency powers and subjective illegibility rating. Previous studies have shown that the illegibility of text varies depending on the combination of sentences and background colors (*Zorko et al., 2017*). Therefore, to vary sentence illegibility, we used the following combination of sentences and background colors: black-white, blue-white, yellow-white, white-black, blue-black, and yellow-black. Black and white colors were selected because black sentences with white backgrounds are popular and the most legible in printed materials as well as in digital displays (*Zorko et al., 2017*). We also selected the blue and yellow colors because yellow and blue sentences with white and black backgrounds, respectively, were empirically illegible.

We hypothesized that using glare monitors results in increased illegibility rating, longer reading time, and longer fixation duration because of the discomfort glare or distraction caused by the reflection. These negative effects were also hypothesized to be prominent when a black background is used because it causes reflections to stand out in the glare monitor screen. Furthermore, these effects are reflected in EEG frequency powers in the frontal or occipital channels.

## MATERIALS AND METHODS

### Participants

Twenty-two (male = 21, female = 1) Japanese undergraduate and graduate students, aged 19–23 years, participated in the experiment. Four were excluded from the analysis because the quality of their EEG (three men) or eye-tracking (one man) data was poor. All participants had normal/corrected-to-normal vision, and those who reported being atypical colorblind were excluded beforehand. This study was approved by the ethics committee of Nagaoka University of Technology (approval number: R3–6) and was conducted according to the Declaration of Helsinki principles. Written informed consent with verbal explanation was obtained from all participants. Participants were verbally informed after the experiment that the purpose of the experiment was to compare the use of a glare monitor with the use of a non-glare monitor.

### Apparatus

For this experiment, a 31.5-inch glare monitor (32MP60G-B, LG) and a 31.5-inch non-glare monitor (JN-IPS315WQHDR, JAPANNEXT) were used. The screen resolution was set to $1,920 \times 1,080$ and the refresh rate was set to 60 Hz for both monitors. We adopted the default settings of brightness and contrast for each monitor, because these monitors are often used as purchased without adjusting the settings. The luminance of black, white, blue, and yellow were approximately 0 cd/m$^2$, 97 cd/m$^2$, 3 cd/m$^2$, and 94 cd/m$^2$ on the glare monitor, respectively, and approximately 0 cd/m$^2$, 194 cd/m$^2$, 6 cd/m$^2$, and 188 cd/m$^2$ on the non-glare monitor, respectively. The height of the monitor screens was adjusted by placing stands under the monitors, so that the participants' faces were reflected about halfway up the monitor screen when using the glare monitor, especially with a black background.

Of the six lights in the room, only the two above the participant's head were turned on during the experiment. A silent video of a first-person shooter (FPS) game (VALORANT, Riot Games, Los Angeles, CA, USA) was projected on the white wall, approximately 5 m behind the participants, using a ceiling-mounted projector. As such, the FPS video game was reflected on the upper part of the monitor (except in areas where the participants' faces were reflected) when using the glare monitor, especially with a black background. The FPS game was selected because it contained a lot of flashing light. The experimenter gave no explanation about the FPS video game to the participants. Brightness around the monitor screen was approximately 300 lux.

An eye tracker (Tobii Pro nano, Tobii AB Stockholm, Sweden) was attached to the lower edge of the monitor screen. Eye-tracking data were acquired using Tobii Pro lab software with a 60 Hz sampling rate. The distance between the monitor and the chair was approximately 57 cm, and the height and position of the chair were adjusted appropriately. The eye tracker settings were properly configured using Tobii Pro Eye Tracker Manager, and eye-tracking calibration was performed using Tobii Pro Lab at the beginning of each experimental session.

Emotiv EPOC X and Emotiv Pro (Emotiv Inc, San Francisco, CA, USA) were used to record EEG data. Emotiv EPOC X is a cost-effective wireless EEG device with 14 electrodes placed at AF3, AF4, F3, F4, F7, F8, FC5, FC6, T7, T8, P7, P8, O1, and O2, based on the international 10-10 system, as well as Driven Right Leg and Common Mode Sense electrodes located at P3 and P4. The sampling rate was set to 256 Hz. The stimulus trigger was sent *via* a serial port from the stimulus presentation program created by PsychoPy (*Peirce et al., 2019*), and EEG data and the trigger were simultaneously recorded using Emotiv Pro software. Emotiv Pro was run on a different PC from the one running Tobii Pro Lab and the stimulus presentation program (Fig. S1). Eye-tracking and EEG data were simultaneously recoded. However, because they were not directly synchronized, stimulus onset was determined by the screen recorded through the video recording function of the eye-tracking software and the serial trigger recorded by EEG data acquisition software when analyzing eye-tracking data and EEG data, respectively.

Although Emotiv EPOC X is a cost-effective EEG device, previous studies have confirmed its reliability (*Barham et al., 2017*; *de Lissa et al., 2015*; *Williams, McArthur & Badcock, 2021*). Emotiv's EEG devices previously had problems of inaccurate trigger timing (*Hairston, 2012*; *Ries et al., 2014*); however, this has been resolved (*Williams, McArthur & Badcock, 2021*).

### Stimuli

The stimuli comprised 180 Japanese sentences with 14 to 39 Japanese characters (average 26.7 characters). The sentences were sampled randomly from Yahoo News Japan (https://news.yahoo.co.jp/) without considering the linguistic structure of the sentences. This was done because the present study did not consider specific aspects of linguistic processing, and instead aimed to examine the difference between using glare and non-glare monitors. Contextually, these sentences were not connected; however, some of them shared the same topic (*e.g.*, COVID-19). The sentences were presented on single lines and the Japanese characters presented in the monitor were at approximately 1.5 degree of visual angle. The combination of sentences (including the fixation point) and background colors were black-white, blue-white, yellow-white, white-black, blue-black, or yellow-black. Each color was set in RGB, as follows: black (0, 0, 0), white (255, 255, 255), blue (0, 0, 255), and yellow (255, 255, 0).

The 180 sentences were divided into 12 blocks (each comprising 15 sentences) because the monitors (glare or non-glare) and sentence/background colors (black-white, blue-white, yellow-white, white-black, blue-black, or yellow-black) factors totaled 12 combinations. On average, the sentences assigned to each block had approximately the same number of characters (26.3 to 27.1 characters). The presentation order of the sentences in the experiment was fixed, and the order of sentence/background colors assigned to each block (15 consecutive trials for each sentence/background colors) was randomized.

## Task

After wearing the Emotiv EPOC X headset, participants sat in chairs in front of a glare monitor or a non-glare monitor, which were side by side in the experiment room. Whether to first sit in front of the non-glare or glare monitor was counterbalanced across participants. Before the onset of the first experimental session, participants performed six practice trials using the assigned monitor. In the practice trials, each color combination was presented once in a random order. The eye tracker was calibrated before the first experimental session started.

In the experimental session, six blocks, each comprising 15 successive trials, were presented (Fig. 1). Each trial comprised a fixation point presented for 1 s, followed by a sentence presented for five seconds. The colors of the sentences (including the fixation point) and background were the same within a block. Participants were instructed to read the sentences silently with their bodies and heads remaining as still as possible. They were also instructed to look at the fixation point when it appeared and press the Enter key on the keyboard after they finished reading a sentence. The stimuli were presented at the lower part of the monitor, where neither the participants' faces or the FPS video game were reflected, even when the glare monitor was used. The horizontal position of the first character was fixed, regardless of the number of characters in a sentence (*i.e.*, centering was not performed). The horizontal coordinate of the fixation point was in the middle of the screen. After the end of each block, the illegibility of the sentences in the block was rated using an eight-point Likert scale, ranging from 1 (legible) to 8 (illegible). During the rating, we presented a red-colored Likert scale in the center of the screen with a pink background. Participants were instructed to try to rate using a wide range of values.

After the first experimental session was completed, the participants sat in front of the other monitor and the eye tracker was calibrated. Then, they began the second experimental session in the same manner using the other monitor. The sessions lasted approximately 10 min.

## Analysis of eye-tracking data

Each gaze data were classified into "Fixation," "Saccade," "Unclassified," or "EyesNotFound" by Tobii Pro Lab's fixation filter, and only "Fixation" data were used for analysis. Three areas of interest (AOI) were defined (Fig. S1), as described herein: where the sentences were presented and its vicinity (referred to as sentence AOI); where the face was reflected, especially when using a glare monitor with a black background (referred to as face AOI); the upper part of the monitor screen where the FPS video game was reflected when a glare monitor was used (referred to as upper AOI). However, the result of the upper AOI was not discussed because of the low number of fixations in this AOI (Table S1).

The mean duration of fixations that occurred in the sentence AOI from stimulus onset to 2,300 ms after were counted in each condition. We determined 2,300 ms was the period end because it was approximately one standard deviation less than the average reading time (as per the reading time analysis result), which meant that the reading was not finished at this point in most cases (approximately 85%).

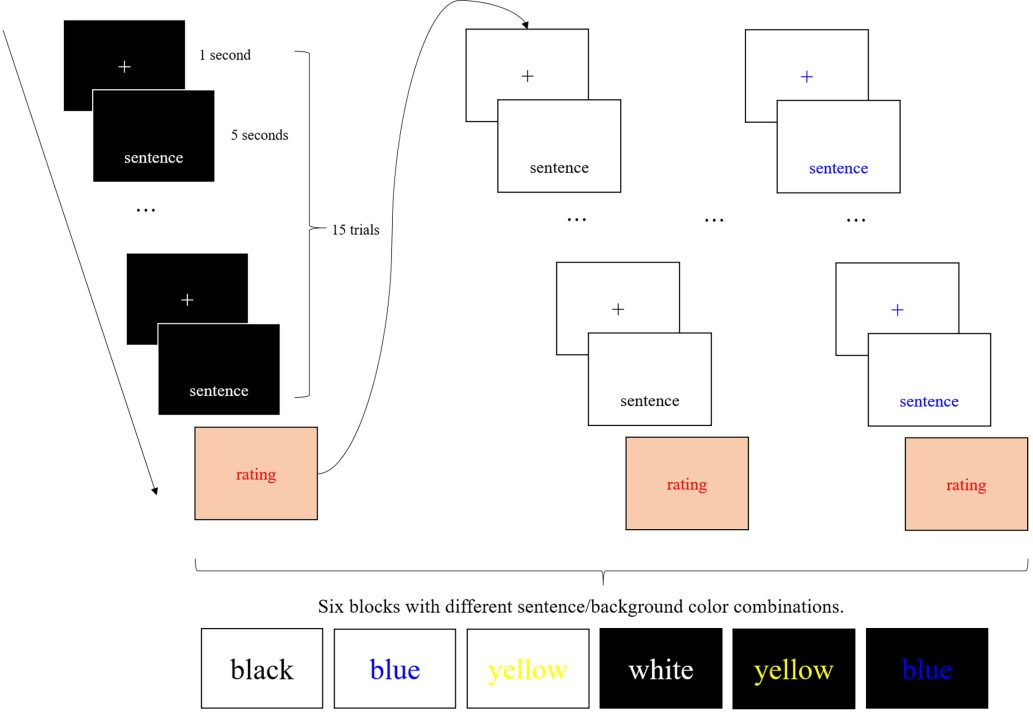

**Figure 1 Schematic illustration of the experimental procedure.** In the experimental session, six blocks, each comprising 15 successive trials, were presented. Each trial comprised a fixation point presented for 1 s, followed by a sentence presented for 5 s. The colors of the sentences and background were the same within a block. After the end of each block, the illegibility of the sentences in the block was rated using an eight-point Likert scale, ranging from 1 (legible) to 8 (illegible).

Further, the number of fixations that occurred in the face AOI from stimulus onset to 5,000 ms after (*i.e.*, during sentence presentation) was counted in each condition. As the fixations in the face AOI were few, we considered the background color as a factor, regardless of sentence color, instead of the combination of sentence/background colors. We prioritized background color over sentence color because the glare monitor shows a noticeable reflection when the background is black.

To compare the difference between conditions, within-participant analysis of variances (ANOVAs) were performed on the duration of fixations in the sentence AOI and the number of fixations in the face AOI.

## Analysis of EEG data

EEGLAB version 2022.1 was used for EEG data analysis. First, a 1 Hz high-pass filter was applied, and then Artifact Subspace Reconstruction (*Chang et al., 2020*) was used to correct for noisy data. Thereafter, EEG data from 1,000 ms before stimulus onset to 5,000 ms after stimulus onset was epoched. The periods between 1,000 ms before stimulus onset to stimulus onset served as the baseline. Independent component analysis was performed, and only "brain" components with a probability of more than 50% confidence, as classified by IClabel (*Pion-Tonachini, Kreutz-Delgado & Makeig, 2019*), were retained. Afterward, visual inspection was performed and a few remaining artifactual components were

excluded. Two participants for whom more than one-third of the trials were unavailable, and one participant for whom more than half of the trials were unavailable in one monitor condition, were excluded from the analysis. The average number of remaining trials was 158.8 (SE = 4.1).

We calculated the grand average event-related spectral perturbation (ERSP) (*Makeig, 1993*) in the F3, F4, O1, and O2 channels in all conditions and for all participants. We focused on these four channels because the frontal and occipital regions are important for the processing of visually presented language stimuli. Based on the results, the frequency and time interval to be used for later comparison between conditions were decided for the theta, alpha, and beta bands. The analysis periods for comparison between conditions were 0 to 1,400 ms after stimulus onset because 1,400 ms was approximately two standard deviations less than the average reading time. Therefore, the reading had not been completed at this point in most cases (approximately 97.5%). The range was narrower than that in for the fixation analysis to include only minimal brain activity related to motor preparation for key pressing. We also limited the upper edge of the beta band frequency to 20 Hz (*i.e.*, low beta) because lower and higher beta activities might contribute with different language processing aspects (*Weiss & Mueller, 2012*), and high beta oscillations (20–30 Hz) might reflect motor and sensory aspects (*Scaltritti, Suitner & Peressotti, 2020*). Further, the mean power changes in each frequency band relative to the baseline were calculated for each condition.

To compare the difference between conditions, within-participant ANOVAs were performed with theta, alpha, and beta powers as dependent variables and monitor, background color, time range (early or late), region (frontal or occipital), and hemisphere (left or right) as independent variables. We also examined Spearman's rank correlation of each participant's illegibility ratings with their theta, alpha, and beta powers for each sentence/background color condition in each monitor.

## RESULTS

### Illegibility rating

Table 1 presents the mean scores of illegibility rating in each condition. Two participants rated the numbers completely reversed; therefore, the data were reversed and included in the analysis. We conducted an ANOVA using Greenhouse–Geisser correction (*Greenhouse & Geisser, 1959*) with monitor and sentence/background color as within-participant factors. Generalized eta squared (*Olejnik & Algina, 2003*) was reported as the effect size. The results showed a significant main effect of sentence/background color (F(5, 85) = 93.841, $\varepsilon$ = 0.74, $\eta_g^2$ = 0.723, $p < 0.001$). The main effect of monitor (F(1, 17) = 0.000, $\eta_g^2$ = 0.000, $p$ = 1.000) and the interaction effect between monitor and sentence/background color (F(5, 85) = 0.886, $\varepsilon$ = 0.76, $\eta_g^2$ = 0.011, $p$ = 0.474) were not significant. Multiple comparisons using the Holm method showed that the yellow-white condition was more illegible than the other conditions (adjusted $ps < 0.001$). The blue-black condition was rated as the second most illegible and was significantly more illegible than the other conditions, except for the yellow-white condition (adjusted

**Table 1 Illegibility rating.**

| Sentence | | Black | Blue | Yellow | White | Blue | Yellow |
|---|---|---|---|---|---|---|---|
| Background | | White | | | Black | | |
| Glare monitor | Mean | 1.94 | 2.50 | 6.89 | 1.67 | 3.89 | 2.50 |
| | SE | 0.24 | 0.29 | 0.30 | 0.16 | 0.32 | 0.29 |
| Non-glare monitor | Mean | 1.78 | 2.56 | 7.17 | 1.78 | 3.44 | 2.67 |
| | SE | 0.22 | 0.28 | 0.22 | 0.17 | 0.35 | 0.32 |

Note:
Scores were provided using an eight-point Likert scale ranging from 1 (legible) to 8 (illegible).

ps < 0.039). The white-black condition was rated as the most legible and significantly more legible than other conditions, except for the black-white condition (adjusted ps < 0.047).

## Reading time

Table 2 presents the mean reading time in each condition. The mean reading time across all conditions was 3,084.8 ms (standard error (SE) = 194.3). Before the mean reading time was calculated, trials that deviated more than three standard deviations from the individual average reading time for all trials were excluded as outliers. Data from three participants who did not press the Enter key at all in at least one condition were excluded from this analysis. These participants were included in other analyses because their eye-tracking data clearly showed that they actually read the sentences.

We conducted an ANOVA using Greenhouse–Geisser correction with monitor and sentence/background color as within-participant factors. The results indicated no significant effect of monitor ($F_{(1, 14)} = 0.134$, $\eta_g^2 = 0.000$, $p = 0.720$), no significant main effect of sentence/background color ($F_{(5, 75)} = 2.473$, $\varepsilon = 0.59$, $\eta_g^2 = 0.009$, $p = 0.076$), and no significant interaction between monitor and sentence/background color ($F_{(5, 75)} = 0.475$, $\varepsilon = 0.69$, $\eta_g^2 = 0.001$, $p = 0.728$).

## Eye-tracking data: fixation duration in the sentence AOI

Table 3 presents the mean fixation duration in the sentence AOI from stimulus onset to 2,300 ms after onset in each condition. We conducted an ANOVA using Greenhouse–Geisser correction with monitor and sentence/background color as within-participant factors. The results indicated the significant effect of monitor ($F_{(1, 17)} = 10.739$, $\eta_g^2 = 0.031$, $p = 0.004$) and the significant main effect of sentence/background color ($F_{(5, 85)} = 10.242$, $\varepsilon = 0.50$, $\eta_g^2 = 0.100$, $p < 0.001$). The interaction between monitor and sentence/background color ($F_{(5, 85)} = 0.742$, $\varepsilon = 0.66$, $\eta_g^2 = 0.005$, $p = 0.544$) was not significant. Multiple comparisons using the Holm method showed that the mean fixation duration of the yellow-white condition was longer than that of the black-white and blue-white conditions (adjusted ps < 0.001).

The mean number of fixations per sentence in the sentence AOI from stimulus onset to 2,300 ms after onset in each condition were also provided in Table S2.

**Table 2  Mean reading time (ms).**

| Sentence | | Black | Blue | Yellow | White | Blue | Yellow |
|---|---|---|---|---|---|---|---|
| Background | | White | | | Black | | |
| Glare monitor | Mean | 2,941.9 | 3,010.0 | 3,206.6 | 3,077.6 | 2,987.9 | 3,032.5 |
| | SE | 179.2 | 188.1 | 197.6 | 186.8 | 228.0 | 218.5 |
| Non-glare monitor | Mean | 2,965.4 | 2,986.4 | 3,139.8 | 2,984.6 | 2,976.9 | 2,989.6 |
| | SE | 210.7 | 205.1 | 218.3 | 197.0 | 223.3 | 226.5 |

Note:
 Neither the main effect of monitor nor the main effects of sentence/background color were significant.

**Table 3  Mean fixation duration in the sentence AOI from 0 to 2,300 ms.**

| Sentence | | Black | Blue | Yellow | White | Blue | Yellow |
|---|---|---|---|---|---|---|---|
| Background | | White | | | Black | | |
| Glare monitor | Mean | 219.9 | 214.9 | 252.7 | 223.1 | 228.6 | 225.6 |
| | SE | 7.7 | 7.8 | 9.6 | 9.0 | 7.1 | 7.7 |
| Non-glare monitor | Mean | 200.8 | 203.9 | 234.3 | 216.1 | 219.0 | 218.1 |
| | SE | 7.9 | 7.9 | 10.5 | 7.3 | 8.2 | 7.2 |

Note:
 The mean duration of the yellow-white condition was longer than that of the black-white and blue-white conditions.
AOI, area of interest.

## Eye-tracking data: number of fixations in the face AOI

Table 4 presents the mean number of fixations per participant in each condition (*i.e.*, per 45 sentences) in the face AOI from stimulus onset to 5,000 ms after onset. We conducted an ANOVA with monitor and background color as within-participant factors. The results showed a significant main effect of monitor ($F(1, 17) = 8.264$, $\eta_g^2 = 0.124$, $p = 0.011$), a significant main effect of background color ($F(1, 17) = 7.009$, $\eta_g^2 = 0.072$, $p = 0.017$), and a significant interaction between monitor and background color ($F(1, 17) = 8.821$, $\eta_g^2 = 0.084$, $p = 0.009$). The results of simple main effect tests showed that the glare monitor condition had more fixations in the face AOI than the non-glare monitor condition did when the background was black ($F(1, 17) = 8.783$, $\eta_g^2 = 0.197$, $p = 0.009$). Further, the black background condition had more fixations in the face AOI than the white background condition did when the glare monitor was used ($F(1, 17) = 8.045$, $\eta_g^2 = 0.147$, $p = 0.011$).

## EEG data

Grand average ERSPs were calculated in all trials for all participants in F3, F4, O1, and O2 channels (Fig. 2). Based on the results, the conditions were compared from 0–400 ms and from 400–1,400 ms for theta (4–8 Hz), alpha (9–14 Hz), and beta (15–20 Hz) bands.

 In each frequency power, we conducted within-participant ANOVAs with monitor, background color, time range, region, and hemisphere as independent factors. For the analysis of theta power, we found significant main effects of time range ($F(1, 17) = 5.144$, $\eta_g^2 = 0.019$, $p = 0.037$) and region ($F(1, 17) = 11.598$, $\eta_g^2 = 0.059$, $p = 0.003$), and a significant

**Table 4 Mean number of fixations per participant (i.e., per 45 sentences) in the face AOI.**

| Background | | White | Black |
|---|---|---|---|
| Glare monitor | Mean | 2.2 | 11.4 |
| | SE | 0.8 | 3.7 |
| Non-glare monitor | Mean | 1.0 | 0.6 |
| | SE | 0.6 | 0.3 |

Note:
 The number of fixations in the face AOI was greater in the glare monitor with the black background condition than in the non-glare monitor with the black background condition, and than in the glare monitor with the white background condition. AOI, area of interest.

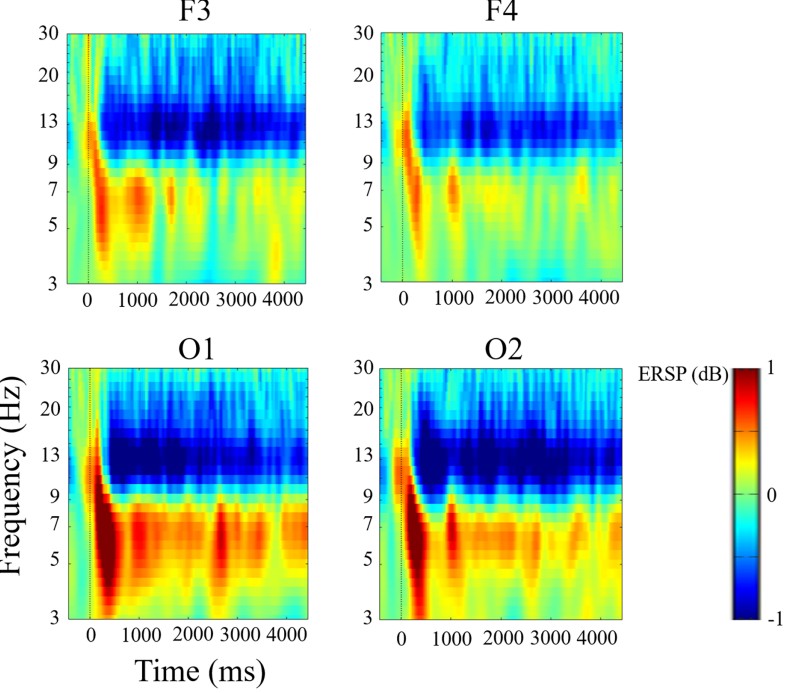

**Figure 2 Grand average event-related spectral perturbation in F3, F4, O1, and O2 channels.** We observed an increase in theta (4–8 Hz) power and a decrease in alpha (9–14 Hz) and beta (15–20 Hz) powers.

interaction between time range and region (F(1, 17) = 5.830, $\eta_g^2$ = 0.005, $p$ = 0.027). These results indicate that the theta power increased more in the occipital channels than in the frontal channels, and the theta power increased more in the early time window than in the later time window in the occipital channels. The main effects of monitor (F(1, 17) = 1.180, $\eta_g^2$ = 0.006, $p$ = 0.293), background color (F(1, 17) = 1.006, $\eta_g^2$ = 0.006, $p$ = 0.330), and hemisphere (F(1, 17) = 3.277, $\eta_g^2$ = 0.010, $p$ = 0.088), as well as other interaction effects (ps > 0.083), were not significant.

For the analysis of alpha power, we found significant main effects of time range (F(1, 17) = 19.192, $\eta_g^2$ = 0.167, $p$ < 0.001) and background color (F(1, 17) = 4.961, $\eta_g^2$ = 0.019, $p$ = 0.040). These results indicate that the alpha power was smaller in the later time windows than in the early time window, and the alpha power decreased less when the

background color was black than when it was white. We also found a significant interaction between time range and region (F(1, 17) = 11.889, $\eta_g^2$ = 0.016, $p$ = 0.003); among time range, region, and background color (F(1, 17) = 6.170, $\eta_g^2$ = 0.003, $p$ = 0.024); among time range, region, and monitor (F(1, 17) = 8.912, $\eta_g^2$ = 0.002, $p$ = 0.008). The main effects of monitor (F(1, 17) = 1.685, $\eta_g^2$ = 0.008, $p$ = 0.212), region (F(1, 17) = 0.145, $\eta_g^2$ = 0.003, $p$ = 0.701), and hemisphere (F(1, 17) = 0.354, $\eta_g^2$ = 0.001 $p$ = 0.560), as well as other interaction effects (ps > 0.102) were not significant. As the interaction between time range, region, and monitor was significant, we conducted within-participant ANOVAs with monitor and region as independent factors in each time window. In the early time window, the results showed a significant main effect of region (F(1, 17) = 8.690, $\eta_g^2$ = 0.071, $p$ = 0.009), indicating that the alpha power increased more in the occipital channels than in the frontal channels. The main effect of monitor (F(1, 17) = 1.526, $\eta_g^2$ = 0.030, $p$ = 0.234) and the interaction between region and monitor (F(1, 17) = 0.104, $\eta_g^2$ = 0.001, $p$ = 0.751) were not significant. In the late time window, the main effects of monitor (F(1, 17) = 0.829, $\eta_g^2$ = 0.007, $p$ = 0.375) and region (F(1, 17) = 2.293, $\eta_g^2$ = 0.011, $p$ = 0.148), and the interaction between region and monitor (F(1, 17) = 3.901, $\eta_g^2$ = 0.006, $p$ = 0.065), were not significant.

For the analysis of beta power, the main effects of monitor (F(1, 17) = 0.824, $\eta_g^2$ = 0.006, $p$ = 0.377), background color (F(1, 17) = 1.117, $\eta_g^2$ = 0.004, $p$ = 0.305), region (F(1, 17) = 3.075, $\eta_g^2$ = 0.009, $p$ = 0.098), and hemisphere (F(1, 17) = 0.657, $\eta_g^2$ = 0.000, $p$ = 0.657) were not significant. The main effect of time range (F(1, 17) = 30.238, $\eta_g^2$ = 0.101, $p$ < 0.001) was significant. The interactions between the following were also significant: time range and region (F(1, 17) = 8.671, $\eta_g^2$ = 0.012, p = 0.009), time range and hemisphere (F(1, 17) = 4.912, $\eta_g^2$ = 0.004, $p$ = 0.041), time range and background color (F(1, 17) = 7.970, $\eta_g^2$ = 0.004, $p$ = 0.012), region and hemisphere (F(1, 17) = 5.524, $\eta_g^2$ = 0.003, $p$ = 0.031), monitor and background color (F(1, 17) = 6.990, $\eta_g^2$ = 0.032, $p$ = 0.017), and time range, hemisphere, and monitor (F(1, 17) = 6.036, $\eta_g^2$ = 0.003, $p$ = 0.025). Of these significant interactions, those involving the monitor factor were investigated further in line with our research interest. The simple main effect test results revealed that beta power decreased less in the glare monitor condition than in the non-glare monitor condition when the background was black (F(1, 17) = 5.192, $\eta_g^2$ = 0.065, $p$ = 0.036). Further, it decreased less in the black background condition than in the white background condition when the glare monitor was used (F(1, 17) = 9.623, $\eta_g^2$ = 0.060, $p$ = 0.007) (Fig. 3). We also conducted within-participant ANOVAs with monitor and hemisphere as independent factors in each time window. However, we found no significant main or interaction effect (ps > 0.131).

## Correlation analysis

For each of the glare and non-glare monitor conditions, we examined the correlation between each participant's illegibility ratings and the theta, alpha, and beta powers in each sentence/background color condition (Table 5). In the glare monitor condition ($n$ = 108), we found that the illegibility rating was significantly negatively correlated with the alpha power in F4 from 400 to 1,400 ms ($r$ = −0.19, $p$ < 0.05); with the beta power in F3 from 400 to 1,400 ms ($r$ = −0.21, $p$ < 0.05) and in F4 from 0 to 400 ms ($r$ = −0.20, $p$ < 0.05).

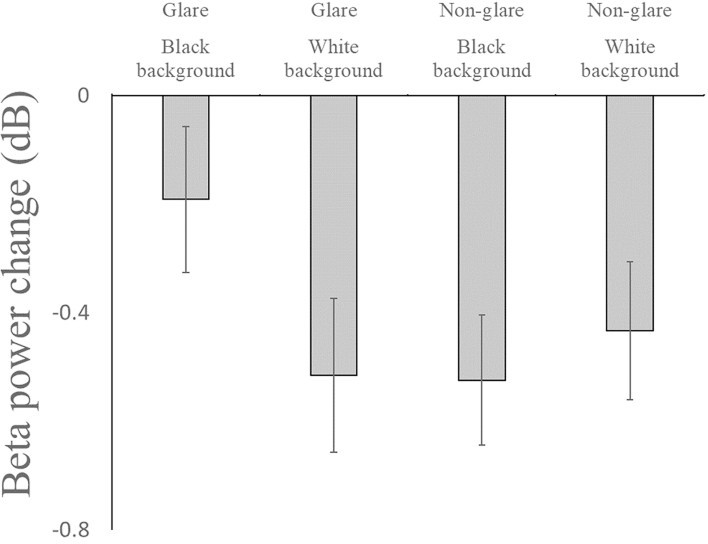

**Figure 3 Beta power change relative to baseline.** The beta power decreased less in the glare monitor with the black background condition than in the non-glare monitor with the black background condition, and than in the glare monitor with the white background condition. Error bars indicate standard errors.                                                                               

**Table 5 Spearman's rank correlation coefficients with illegibility ratings.**

| Monitor | | Glare (*n* = 108) | | Non-glare (*n* = 107) | |
|---|---|---|---|---|---|
| Time | | 0–400 ms | 400–1,400 ms | 0–400 ms | 400–1,400 ms |
| Theta (4–8 Hz) | F3 | −0.07 | 0.04 | 0.17 | 0.24* |
| | F4 | −0.12 | −0.15 | 0.15 | 0.23* |
| | O1 | −0.01 | 0.04 | −0.02 | 0.18 |
| | O2 | −0.12 | −0.13 | 0.03 | 0.14 |
| Alpha (9–14 Hz) | F3 | 0.03 | −0.12 | −0.02 | 0.09 |
| | F4 | 0.01 | −0.19* | 0.11 | 0.05 |
| | O1 | 0.08 | −0.09 | −0.03 | −0.06 |
| | O2 | −0.02 | −0.12 | 0.12 | 0.08 |
| Beta (15–20 Hz) | F3 | −0.10 | −0.21* | 0.10 | 0.17 |
| | F4 | −0.20* | −0.15 | 0.16 | 0.14 |
| | O1 | 0.05 | −0.03 | 0.06 | 0.08 |
| | O2 | −0.18 | −0.12 | 0.25* | 0.15 |

**Note:**
An asterisk (*) significant at the 0.05 level (two-tailed).

In contrast, in the non-glare monitor condition (*n* = 107), the illegibility rating was significantly positively correlated with the theta power in F3 from 400 to 1,400 ms (*r* = 0.24, *p* < 0.05) and in F4 from 400 to 1,400 ms (*r* = 0.23, *p* < 0.05); with the alpha power in O2 from 0 to 400 ms (*r* = 0.25, *p* < 0.05).

## DISCUSSION

We investigated whether the possible negative effects of using a glare monitor during sentence reading could be measured by objective physiological responses using EEG or an eye tracker. The grand average ERSP revealed that theta power increased and alpha power decreased, which was a reliable observed pattern in various cognitive tasks (*Zhu, Wang & Zhang, 2021*). Although no significant effect of monitor on illegibility rating or reading time was found, the fixation duration in the sentence AOI measured using an eye tracker was significantly longer in the glare than in the non-glare monitor condition. Eye-tracking data also indicated that the number of fixations outside the sentence AOI was often higher when using the glare monitor with a black background (*vs* the other conditions). Consistent with this, EEG beta power decreased less when using the glare monitor with black background than in other conditions. The main findings are summarized in Table 6.

The lack of a significant effect of monitor on illegibility rating or reading time was unexpected. Based on the finding of a longer fixation duration when a glare monitor was used (*vs* non-glare monitor), it may be that the settings of the current experiment hindered the detection of subtle differences in the illegibility rating or reading time between the glare and non-glare monitor conditions. One possible reason for the lack of effect on illegibility rating is that the participants might have tried to rate the difference in illegibility between the combination of background and sentence colors within each monitor condition by using various values, rather than to compare illegibility in relation to the use of a glare monitor and that of a non-glare monitor. The very large difference in illegibility between the yellow sentence with white background and other color combinations may have masked the perceived illegibility difference between the glare and non-glare monitors. Regarding the possible reason for the lack of a significant effect of monitor on reading time, the measurement of reaction time based on keystrokes is more likely to evoke more variation than the measurement of fixation duration; this is because the former requires a conscious decision and the latter is subconsciously controlled. Taken together, we interpreted that the illegibility caused by glare monitor use was subtly but certainly present and could only be observed in fixations with a longer duration.

Consistent with our hypothesis, eye-tracking data revealed that the fixations outside the sentence AOI occurred especially when a glare monitor with a black background was used —that is, when the participants' faces and the FPS video game were clearly reflected on the monitor. Furthermore, most of the fixation that occurred outside the sentence AOI was in the face AOI rather than in the upper AOI, suggesting that the face reflected on the glare monitor was more distracting than the reflection of the FPS video game. This may be because the self-face automatically captures attention because of its importance (*Bola et al., 2021*; *Jublie & Kumar, 2021*).

The ANOVA results showed that the EEG beta power decreased less when using the glare monitor with a black background than in other conditions. Although beta band activity has been associated with sensory motor processes (*Pfurtscheller & Lopes da Silva, 1999*), beta oscillations are involved in various cognitive functions (*Weiss & Mueller, 2012*). Thus, they are now considered as related to the maintenance of the current perceptual or

**Table 6 Summary of the main findings of the statistical analyses.**

| Significant main effect of monitor | |
| --- | --- |
| Fixation duration in the sentence AOI | *Glare (227.5 ms) > non-glare (215.4 ms)* |
| Number of fixations in the face AOI | There was also a significant interaction between monitor and background color. |
| **Significant interaction including monitor** | |
| Number of fixations in the face AOI | *(Monitor * background color)* <br> *Glare-monitor with black background > glare monitor with white background, non-glare monitor with black background.* |
| Alpha power (9–14 Hz) | (Monitor * time range * region) <br> No significant effect of monitor in the follow-up analysis. |
| Beta power (15–20 Hz) | *(Monitor * background color)* <br> *Glare-monitor with black background > glare monitor with white background, non-glare monitor with black background.* <br> (Monitor * time range * hemisphere) <br> No significant effect of monitor in the follow-up analysis. |

**Note:**
The significant main effects or interactions of monitor are listed. AOI, area of interest.

cognitive state (*Engel & Fries, 2010*), and endogenous (re)activation of a cognitive state (*Spitzer & Haegens, 2017*). The decrease in beta band power is typically observed when participants respond in a bottom-up manner because of unexpected exogenous stimuli, as the current cognitive state is disrupted (*Engel & Fries, 2010*). Therefore, we interpreted that the use of a glare monitor with a black background made participants less engaged in the task. As the number of fixations on the face AOI was relatively small, the result seemed to be caused not only by looking away from sentences but also by the distraction caused by the reflection on the monitor when reading the sentences. Although beta power decreases during preparation and execution of movements, the difference in beta power among conditions in our study is likely to not have been caused by the movement-related brain activity; this is because the reading time measured by pressing a key was not significantly different between conditions. In addition, the time and frequency ranges were limited to include as little movement-related brain activity as possible.

Our results demonstrated different correlation patterns between illegibility and the combination of text and background colors under different monitor conditions. Still, these correlation results should not be associated with the negative effect of using a glare monitor, since the monitor showed no effect on illegibility rating—as aforementioned. When using a glare monitor, beta powers were significantly correlated with the illegibility rating in the F3 and F4 channels. Interestingly, the direction of the correlation was negative, indicating that beta powers decreased more when the sentences were more illegible. This finding apparently contradicts the ANOVA result showing that beta powers decreased less when there were distractions in the glare monitor with a black background condition. However, these results are not contradictory. This is because the beta power decrease indexed the disruption of the current cognitive or perceptual state (*Engel & Fries, 2010*; *Spitzer & Haegens, 2017*), not the required cognitive effort or cortical activation. This

description is corroborated by past research showing mixed findings regarding the direction of the correlation between beta band activity and blood-oxygen-level dependent signal (*Michels et al., 2010*; *Hanslmayr et al., 2011*). As such, the beta power decreased less when using a glare monitor with a black background because the engagement in the task or the change in the cognitive state was insufficient owing to the distraction by the reflection in the monitor. The beta power decreased more when the sentences were presented with a more illegible combination of text and background colors because the high demand on perceptual processing for illegible sentence was sufficient to change the perceptual or cognitive state. We also found that the alpha power in the F4 channel was negatively correlated with illegibility rating. Alpha band activity is related to the top-down inhibitory control process or disengagement of task-irrelevant brain regions, and decreased alpha power has been associated with cognitive effort (*Klimesch, 2012*; *Jensen, Bonnefond & VanRullen, 2012*). Conversely, higher alpha power was associated with internally directed cognition (*Ceh et al., 2020*). Therefore, this result indicates that more cognitive effort was required when the sentence was illegible.

In the non-glare monitor condition, we found that theta powers in F3 and F4 were positively correlated with the illegibility score. The theta band activity has been associated with the prioritization of working memory representations (*de Vries, Slagter & Olivers, 2020*; *Riddle et al., 2020*), and the power of the front medial theta increases with increased working memory demand (*Ratcliffe, Shapiro & Staresina, 2022*). Thus, the results indicate that working memory demand increased when a sentence was more illegible. We also found that the beta power in the visual areas were positively correlated with the illegibility rating in the early time window, suggesting that less adjustment was observed in the visual areas to read the more illegible text. This result contradicts our findings for frontal channels, but could be interpreted consistently if participants relied more on working memory than visual processing when illegible sentences were shown on a non-glare monitor. Taken together with the correlation results of the glare monitor condition, we interpreted that more adjustment was required in the frontal brain areas to read the more illegible text. This occurred both when the glare and non-glare monitors were used.

Still, the reasoning remains unclear as to why significant positive correlations between illegibility rating and theta power were not observed in the glare monitor condition, and why significant negative correlations between illegibility rating and alpha or beta powers were not observed in the non-glare monitor conditions. Although theta power increase and alpha power decrease are typically observed when cognitive effort increases (*Zhu, Wang & Zhang, 2021*), theta power increase has been associated more with working memory (*de Vries, Slagter & Olivers, 2020*; *Riddle et al., 2020*; *Ratcliffe, Shapiro & Staresina, 2022*; *Maurer et al., 2015*; *Klimesch, 1999*), whereas alpha power decrease has been associated more with visuospatial attention (*Lobier, Palva & Palva, 2018*; *Capotosto et al., 2009*; *Sauseng et al., 2011*). Thus, we speculate that the illegibility of sentences seen in a non-glare monitor might be related to the cognitive load associated with sentence comprehension, whereas the illegibility of sentences seen in a glare monitor condition

might be related to the perceptual demand associated with the visual decoding of sentences.

This study was not designed to separate the effects of discomfort glare from the distraction caused by the reflection in the glare monitor. Rather, we aimed to examine whether the possible negative effects of using a glare monitor when reading could be measured based on objective physiological responses measured using EEG or an eye tracker. From an engineering perspective, our results are encouraging as they demonstrate that cost-effective EEG tools could measure the subtle effects in an objective manner and can be used for the ergonomic evaluation of products.

There are some limitations in this study. First, the luminance differed between the glare and non-glare monitors. However, this does not explain the observed interaction when using a glare monitor with a black background. A previous study also reported that the colors of the stimulus and background did not influence the EEG frequency band data (*Tian et al., 2022*). Second, we performed extensive statistical analysis because of the exploratory nature of this study; this increased the risk of type 1 error. Third, study participants were mostly limited to young men. Therefore, it is important to conduct investigations in a wider population, particularly older adults. It has been suggested that the effects of discomfort glare vary with age (*Wolska & Sawicki, 2014*); particularly, older adults rated the degree of subjective discomfort glare lower than young adults, but their objective measures of fatigue were greatly influenced by discomfort glare. It would be worthwhile to investigate how physiological indicators during glare monitor use associate with subsequent fatigue.

## CONCLUSIONS

This study examined whether the possible negative effects of using a glare monitor during reading could be measured based on objective physiological responses using an eye tracker and EEG. Although the findings for illegibility rating and reading time did not show the significant effect of monitor type, eye-tracking data indicated that the duration of fixations was longer when using a glare monitor than when using a non-glare monitor. Further, when the background was black, the participants would look away from the sentence AOI and see their faces reflected in the glare monitor, although this occurred few times. The EEG data also revealed that beta power decreased less when using the glare monitor with a black background than in other conditions, suggesting that the participants were less engaged in the reading task in this combination of monitor and background condition. These results indicate that physiological measures such as EEG and eye tracking can be used to measure subtle negative effects of glare monitor use, even if the behavioral measures (*e.g.*, subjective illegibility ratings or reading time) do not show relevant differences. In future work, researchers could confirm the reproducibility and generalizability of our findings, as well as clarify the rationale behind the differences in the correlations between illegibility rating and brain activity under different monitor conditions (*e.g.*, using a glare monitor and a non-glare monitor).

### Funding

This work was supported by JSPS KAKENHI Grant Number 22K02885. The funders had no role in study design, data collection and analysis, decision to publish, or preparation of the manuscript.

### Grant Disclosures

The following grant information was disclosed by the authors:
JSPS KAKENHI: 22K02885.

### Competing Interests

The authors declare that they have no competing interests.

### Author Contributions

- Yoritaka Akimoto conceived and designed the experiments, analyzed the data, prepared figures and/or tables, authored or reviewed drafts of the article, and approved the final draft.
- Keito Miyake conceived and designed the experiments, performed the experiments, analyzed the data, authored or reviewed drafts of the article, and approved the final draft.

### Human Ethics

The following information was supplied relating to ethical approvals (*i.e.*, approving body and any reference numbers):

This study was approved by the ethics committee of Nagaoka University of Technology.

### Data Availability

All raw data used in the statistical analysis are available in the Supplemental Files.

### Supplemental Information

Supplemental information for this article can be found online at http://dx.doi.org/10.7717/peerj.15992#supplemental-information.

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
