# Peer review of "Examination of distraction and discomfort caused by using glare monitors: a simultaneous electroencephalography and eye-tracking study"

_PeerJ, doi:10.7717/peerj.15992_

## Round 0.1 · original submission · Major Revisions

Dear Authors,

There are statistical methodology queries as per the 3 peer reviewers. Please revise and redo as per comments.

Reviewer 1 ·

Basic reporting

- The authors provided a clear and comprehensive description of the study design and outlined the experiment process. The procedure for setting up the experiment was articulated in a meticulous manner and detailed information about the methods used were included.

- In the results section, numerous analysis results (ANOVA) were presented within the text, often in sentence form. While this provides detailed information, it can make it challenging for readers to grasp the overall picture and make comparisons across different analyses. It will be helpful to summarize the key findings in different sections in well-structured and concise tables.

Experimental design

- In examining the correlation between EEG frequency powers and subjective illegibility rating, the authors didn’t specify the type of correlation used in the analysis. It is important to explicitly state whether you employed Pearson correlation or Spearman correlation in order to ensure the accuracy and reproducibility of your findings. To analyze Likert scale variables, non-parametric correlation measures such as Spearman's rank correlation are usually used. These methods are specifically designed to handle ordinal data and do not assume linearity or specific distributional properties.

- One main goal of this study is to find objective physiological responses measured using EEG or an eye tracker to examine the negative effects of using a glare monitor. However, the alpha, beta and theta power in different channels behave differently in the glare and non-glare monitor conditions. Then how to apply these signals to compare the different effects across glare and non-glare monitors instead of evaluating the negative effects within one single condition?

Validity of the findings

- Given that the study population consists solely of undergraduate and graduate students and most of them are male, the study findings may lack generalizability and external validity. This limitation may hinder the broader applicability of your results to a more diverse population. More discussion can be added in the paper to discuss the limitations of the study in terms of sample representativeness and suggest avenues for future research that address these limitations.

- In the illegibility rating results section, the main effect of the monitor has a p value equals 1. Is there any reason behind this unexpectedly high p-value?

Reviewer 2 ·

Basic reporting

1. Generally, the manuscript is very well written. there is no issues in the language use throughout.
2. sufficient amount of information provided in the literature and the data are well structured.
3. However, there is lack of information in the figure captions, and i would like to suggest additional figure to be included especially in the method section.

Experimental design

The methodology section need further clarification:

1. Does the eye tracking and EEG been recorded simultaneously?
2. If it is so, are those data from both modalities been synchronised?
3. it is recommended to add a figure to illustrate the experimental procedure, in order to assist reader understanding on how the experiment been conducted.

Validity of the findings

no comment

Additional comments

Lacking information in all of the figure captions.

Reviewer 3 ·

Basic reporting

The study by Yoritaka Akimoto and Keito Miyake showed the negative effect of using a glare
monitor. While there are some interesting finding, they should be organise the structure of the paper well and be clear when presenting their study.

1,For the result part, each result should have a conclusion. For example, "Illegibility rating", "Reading time", "EEG data" are not a conclusion. They need to conclude what is the findings, so that readers can follow them easily.

2, they also need to give a background in each experiment, for example, the purpose of each experiment

3, they should discuss some result more carefully, especially when data are not controversy.

Experimental design

no

Validity of the findings

no

Additional comments

no

---

## Round 0.2 · accepted · Accept

Thank you for your revised manuscript has been accepted.

Reviewer 1 ·

Basic reporting

I think that the authors have adequately addressed the comments made by the reviewers in the revised version of the manuscript. Therefore, I have no further comments.

Experimental design

I think that the authors have adequately addressed the comments made by the reviewers in the revised version of the manuscript. Therefore, I have no further comments.

Validity of the findings

I think that the authors have adequately addressed the comments made by the reviewers in the revised version of the manuscript. Therefore, I have no further comments.

Reviewer 2 ·

Basic reporting

no comment

Experimental design

All the comments have been addressed.

Validity of the findings

no comment

Additional comments

no comment